# Percutaneous Endoscopic Cervical Discectomy versus Anterior Cervical Discectomy and Fusion: A Comparative Cohort Study with a Five-Year Follow-Up

**DOI:** 10.3390/jcm9020371

**Published:** 2020-01-29

**Authors:** Yong Ahn, Han Joong Keum, Sang Ha Shin

**Affiliations:** 1Department of Neurosurgery, Gil Medical Center, Gachon University College of Medicine, Incheon 21565, Korea; 2Department of Neurosurgery, Wooridul Spine Hospital, Seoul 06068, Korea; hanjoongkeum@hotmail.com (H.J.K.); anconeus@daum.net (S.H.S.)

**Keywords:** anterior cervical, cervical discectomy, endoscopic, hospital stay, operative time, percutaneous, return to work

## Abstract

Percutaneous endoscopic cervical discectomy (PECD) is an effective minimally invasive surgery for soft cervical disc herniation in properly selected cases. The current gold standard is anterior cervical discectomy and fusion (ACDF). However, few studies have evaluated the outcome of PECD compared with ACDF. We compared the surgical results of PECD and ACDF. Data from patients treated with single-level PECD (*n* = 51) or ACDF (*n* = 64) were analyzed. Patients were prospectively entered into the clinical database and their records were retrospectively reviewed. Perioperative data and clinical outcomes were evaluated using the visual analogue scale (VAS), Neck Disability Index (NDI), and modified Macnab criteria. VAS and NDI results significantly improved in both groups. The rates of excellent or good results were 88.24% and 90.63% in the PECD and ACDF group, respectively. The revision rates were 3.92% and 1.56% in the PECD and ACDF group, respectively. Operative time, hospital stay, and time to return to work were reduced in the PECD group compared to the ACDF group (*p* < 0.001). The five-year outcomes of PECD were comparable to those of conventional ACDF. PECD provided the typical benefits of minimally invasive surgery and may be an effective alternative for treating soft cervical disc herniation.

## 1. Introduction

Currently, anterior cervical discectomy and fusion (ACDF) is regarded as the gold standard surgical option for cervical disc herniation (CDH). However, this technique can be accompanied by considerable approach- [1,2,3,4] and fusion-related complications [5,6,7]. Therefore, a minimally invasive surgical option using the percutaneous endoscopic approach was developed. Since Hijikata [8] and Kambin [9] first reported the percutaneous discectomy technique, minimally invasive or endoscopic discectomy techniques have been developed in lumbosacral, cervical, and thoracic spine surgery. For the cervical spine, various percutaneous endoscopic cervical discectomy (PECD) techniques have evolved in selected cases [10,11]. The PECD prototype is fluoroscopically guided percutaneous cervical disc decompression without endoscopic visualization, such as automated nucleotomy [12,13], percutaneous cervical disc chemonucleolysis [14], and percutaneous cervical laser disc decompression [15,16,17]. Since the early 2000s, increasingly practical PECD techniques have been introduced because of advancements in working channel endoscope and surgical instrument technology [18,19,20,21]. However, few studies have assessed the efficacy of this novel technique. To the best of our knowledge, only one randomized controlled trial has been conducted to determine the efficacy of PECD [22]. Furthermore, a paucity of cohort studies has compared the PECD technique and conventional ACDF over a long-term follow-up period. In addition, it is unknown whether PECD exhibits the typical properties of minimally invasive spine surgery (MISS) when compared with open surgery. The purposes of this study were to demonstrate the long-term outcomes of PECD compared with ACDF and to evaluate the invasiveness of this endoscopic technique for soft CDH.

## 2. Materials and Methods

### 2.1. Patient Population

This study was approved by the institutional review board of Wooridul Spine Hospital (approval number WRDIRB 2016-12-011), and written informed consent was obtained. Between January 2009 and September 2011, 135 consecutive patients with symptomatic soft CDH underwent PECD or ACDF. Inclusion criteria for surgery were (1) the presence of a single-level, soft CDH detected by MRI and CT; (2) cervical radicular pain and signs compatible with the radiographic findings; and (3) patients who failed non-operative treatments for ≥6 weeks. Patients with calcified disc herniation, cervical spondylolisthesis, severe cervical spondylotic myelopathy, and advanced spondylosis with narrowed disc space were excluded. Patients were prospectively entered into the clinical database and their records were retrospectively reviewed. Twenty patients (14.8%) were lost to follow-up. Data of the remaining 115 patients were obtained at each follow-up points.

For the same criteria, the operation method was chosen based on the surgeon’s preference. Of the 115 patients, 51 were treated with PECD by expert endoscopic surgeons (PECD group), and the remaining 64 were treated with standard ACDF by surgeons who preferred conventional surgery (ACDF group). The ease or feasibility of the surgical technique did not play a role in the selection of surgery.

### 2.2. Outcome Measurements

Five-year data were obtained from regular outpatient visits by a chart review and patient-based outcome questionnaire. Clinical outcomes were evaluated using the visual analogue scale (VAS) and Neck Disability Index (NDI) [23]. At the final follow-up, global outcomes were classified into excellent (free of pain, no restriction of activity), good (occasional non-radicular pain, relief of presenting symptoms), fair (improved functional capacity, but still handicapped), or poor (insufficient improvement, further operative intervention required) based on the modified Macnab criteria [24] by an independent researcher. Perioperative data, including the operative time, hospital stay, and time to return to work, were documented. Return to work was referred to as resuming work tasks/work hours after a period of sick leave [25,26]. Patients were asked when they could return to their ordinary work postoperatively. Cases of postoperative complications and reoperation were also documented.

### 2.3. Surgical Techniques

PECD was conducted according to a technique that was described previously (Figure 1) [20,21]. Briefly, placed in the supine position, with the neck mildly extended, the patient was monitored under conscious sedation. An anterior percutaneous approach was performed under fluoroscopic control. A contralateral approach was typically used because it provides a better surgical field for the herniated disc. An 18-gauge needle was inserted into the disc through a safe working space between the carotid artery and trachea. Then intraoperative discography was conducted with a mixture of indigo carmine plus contrast dye to stain the nucleus. After adequate sequential dilation process, a final working sheath, 4 mm in diameter, was placed in the disc, with the end of the working sheath on the posterior vertebral line. Next, a working channel endoscope was introduced and anatomical structures, such as the disc, annular anchorage, posterior longitudinal ligament, endplate, and osteophytes, were inspected. Finally, the herniated disc was selectively removed using endoscopic forceps, radiofrequency, and a side-firing holmium yttrium-aluminum-garnet laser. The central nucleus was preserved to minimize postoperative disc collapse or kyphosis, whereas the posterior part of the disc was decompressed. The herniated disc can be taken out after the release of the annular anchorage. Selective discectomy continued via repetition of this “release and remove” maneuver. The released hernia fragment was gradually removed, and epidural pulsation was visualized. The procedure endpoint was the free mobilization and strong pulsation of the thecal sac and nerve root.

Conventional ACDF was performed using the standard technique with a microscope [22]. Briefly, a transverse skin incision was made, and the routine anterior cervical approach was used. After adequate microsurgical cervical discectomy and endplate preparation, a polyether ether ketone (PEEK) cage with bone chips was inserted under fluoroscopic guidance. There was no additional placement of an anterior plate and screw system. The wound was closed layer by layer with drainage.

It was recommended for all patients to keep a neck brace for six weeks postoperatively.

### 2.4. Statistical Analysis

Statistical analysis was performed by an independent researcher using SPSS 14.0K (SPSS, Inc., Chicago, IL, USA). A comparison of continuous variables between the two groups was performed using independent *t*-tests. Fisher’s exact test was used to compare categorical variables. A *p* value < 0.05 was considered statistically significant.

## 3. Results

### 3.1. Demographics and Clinical Outcomes

In the PECD group, the mean age of the 51 patients was 44.2 years (range, 25–67 years). In the ACDF group, the mean age of the 64 patients was 47.5 years (range, 20–75 years). There were no differences in demographic characteristics between the two groups (Table 1). The mean VAS scores for neck pain improved from 4.58 ± 1.95 to 1.35 ± 1.34 in the PECD group and from 3.91 ± 1.78 to 1.14 ± 0.85 in the ACDF group (Figure 2A). The mean VAS scores for radicular pain improved from 7.16 ± 1.78 to 1.61 ± 1.34 in the PECD group and from 7.61 ± 1.27 to 1.44 ± 1.08 in the ACDF group (Figure 2B). The NDI improved from 51.87 ± 21.47 to 7.82 ± 13.41 in the PECD group and from 58.27 ± 17.73 to 6.59 ± 10.14 in the ACDF group (Figure 3). Based on modified Macnab criteria, an excellent or good outcome was achieved in 45 cases (88.24%) of the PECD group and in 58 cases (90.63%) of the ACDF group (Figure 4). However, there were no differences in these clinical outcomes between the PECD and ACDF groups.

### 3.2. Perioperative Data

The PECD group showed a shorter operative time, hospital stay, and time to return to work compared with the ACDF group. Mean operative times were 55.20 ± 18.03 min in the PECD group and 124.53 ± 35.68 min in the ACDF group (*p* < 0.001, Figure 5A). Mean hospital stays were 2.18 ± 1.16 days in the PECD group and 5.23 ± 2.93 days in the ACDF group (*p* < 0.001, Figure 5B). Mean times to return to work were 3.14 ± 1.08 weeks in the PECD group and 10.84 ± 3.12 weeks in the ACDF group (*p* < 0.001, Figure 5C).

### 3.3. Complications and Revisions

The most common complication was swallowing difficulty. Three patients in the ACDF group and one in the PECD group complained of transient swallowing difficulties, which improved within one month postoperatively. Two surface hematomas with mild discomfort were recorded in the ACDF group. No further major complications, such as epidural hematoma, wound infection, neural injury, hoarseness, or postoperative instability, occurred.

The overall revision rate was 2.6% (3 of 115 patients). Two patients in the PECD group underwent subsequent ACDF because of recurrent disc herniation within two months. One patient in the ACDF group underwent subsequent posterior cervical foraminotomy because of postoperative foraminal stenosis 12 months later. There were no differences in the complication and recurrence rates between the groups.

## 4. Discussion

Our data revealed the benefits of MISS in the PECD group in several ways. First, patients who underwent PECD had a shorter operative time than those who underwent ACDF, which may be associated with reduced tissue trauma. Second, having shorter hospital stays and time to return to work indicated less postoperative disability and better quality of life. This may attribute to better cost-effectiveness. Last, a small stab wound without skin incision can provide an excellent cosmetic effect.

Various clinical outcomes were significantly identical between the PECD and ACDF groups. Postoperative changes in pain score and functional status revealed some common patterns over time. Neck and radicular pain scores improved until one year postoperatively, and then slightly increased and stabilized. This phenomenon is more definitive in axial neck pain. We hypothesize that the progression in degenerative change over time may result in this phenomenon. Casal-Moro and colleagues [27] suggested that postoperative degenerative changes increase lumbar or radicular pain after lumbar discectomy during the long-term follow-up.

There are several advantages of PECD as a MISS [11]. First, an anterior percutaneous cervical approach with only a thin working channel endoscope of 3–4 mm diameter can minimize normal tissue trauma. Therefore, there is a reduction in scarring and faster rehabilitation when compared with open surgery. Moreover, unnecessary instrumentation can be prevented while preserving segmental stability. Second, selective removal of a herniated disc fragment is performed while preserving the maternal disc. Disc herniation at any zone, from the central to the extraforaminal, and any degree of disc herniation, from an annular tear to extrusion, can be effectively removed under delicate endoscopic visualization. Finally, PECD can be performed under either local or general anesthesia, a choice that was made considering the patient’s condition.

However, there are some limitations to this procedure. The spine surgeon could be technically unfamiliar to PECD, which has a long learning curve. In addition, its definitive indication is confined to soft disc herniation. In cases of narrowed disc space, bony stenosis, or hard disc, this procedure is not appropriate or reliable. In practice, there are variable cases of soft cervical disc herniation mixed with a hard component to some extent. Therefore, the reported success rates of PECD have been variable, from 51%–95% [11].

Scientific evidence for this technique has not been fully elucidated. Few randomized controlled trials (RCTs) have evaluated its effectiveness. Ruetten et al. [22] compared the clinical and radiological results of full-endoscopic cervical discectomy (54 cases) and conventional ACDF (49 cases) over a two-year follow-up period. They concluded that full-endoscopic cervical discectomy is a sufficient and safe alternative in selected cases. However, most reviews of PECD have been narrative literature reviews or technical considerations that lack meta-analysis or systematic review [11,28,29]. High-quality RCTs and meta-analyses are required to assess the full relevance and reliability of PECD compared with the conventional ACDF.

Some essential technical points should be considered to obtain consistent and reproducible results using this technique. A target-oriented percutaneous anterior approach is the first key to successful surgery. The approach needle should be inserted contralaterally through the safe working zone, which is the intermuscular zone between the carotid sheath and trachea. This working zone can be made with the surgeon’s fingertips under fluoroscopic control. The tracheal air shadow or endotracheal tube may be a useful mark for a safe working zone. The trajectory of the approach should be directed to the disc herniation point or annular tear site. Adequate annular release before the removal of the herniated fragments is the second key to success. Any attempt to remove the herniated disc fragment without performing the release process cannot achieve a sufficient decompression effect because the endoscopic instruments are smaller and weaker compared to those used for open surgery. Precise determination of the endpoint of the surgery is the final key to success. The minimal condition of a proper endpoint is free pulsation and mobilization of the thecal sac and nerve root, regardless of myelin or dural membrane exposure under endoscopic visualization.

To the best of our knowledge, this is the first study to compare long-term outcomes between PECD and conventional ACDF, with a minimum five-year follow-up. However, the study has some limitations. First, there may be considerable selection bias in the enrollment criteria because the patient selection was not randomized. Second, this study lacked an evaluation of the medical cost and radiological changes over the long-term follow-up. Thus, our next study will be an evaluation of these issues, including the cost-effectiveness, degenerative changes, segmental stability, mobility, and cervical alignment between PECD and conventional ACDF.

## 5. Conclusions

The long-term outcomes of PECD for soft CDH are comparable with those of conventional ACDF. PECD could offer the typical merits of MISS, including reduced operative time, hospital stay, and time to return to work. Taken together, PECD may be an efficient surgical alternative for treating cervical soft disc herniation.

## Figures and Tables

**Figure 1 jcm-09-00371-f001:**
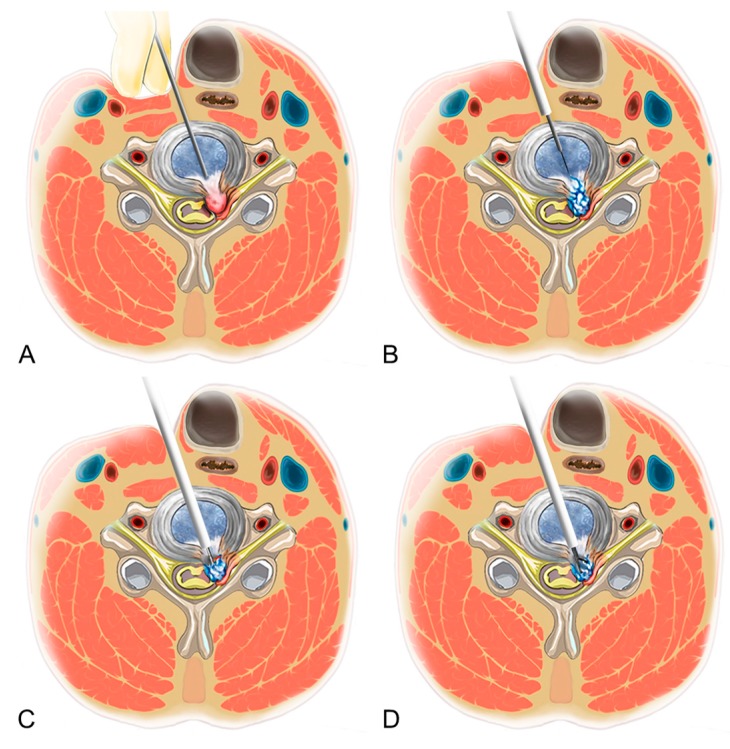
Basic principle of percutaneous endoscopic cervical discectomy (PECD). **A**. Anterior percutaneous cervical approach. The trachea is pushed medially by finger pressure, keeping the carotid sheath laterally. **B**. Sequential dilation technique from the needle approach to the final working sheath via the safe working zone was conducted. **C**. Release of the annular anchorage using the side-firing laser or radiofrequency dissector. **D**. Selective cervical discectomy is performed under direct endoscopic visualization, preserving the central nucleus.

**Figure 2 jcm-09-00371-f002:**
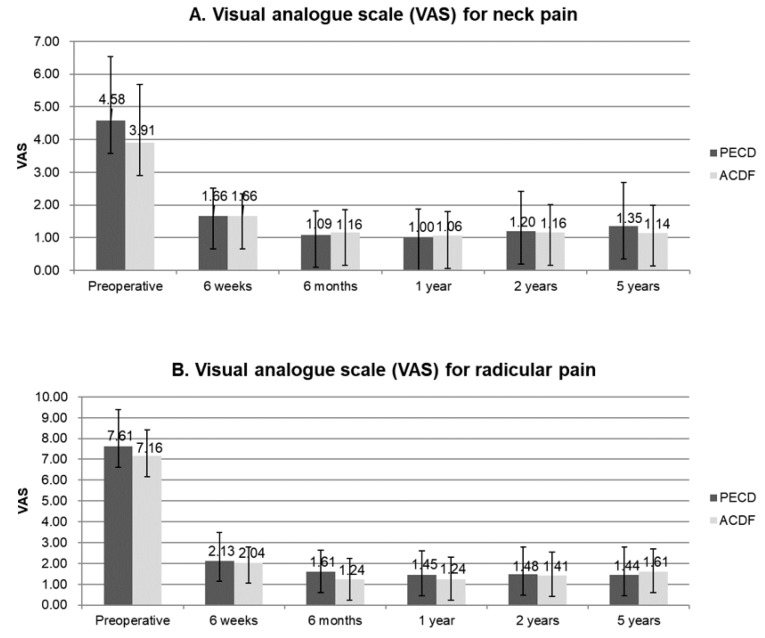
Visual analogue scale (VAS) measured preoperatively and at six weeks, six months, one year, two years, and five years postoperatively. **A.** VAS for neck pain. **B.** VAS for radicular pain. There were no significant differences between the groups. PECD, percutaneous endoscopic cervical discectomy; ACDF, anterior cervical discectomy and fusion.

**Figure 3 jcm-09-00371-f003:**
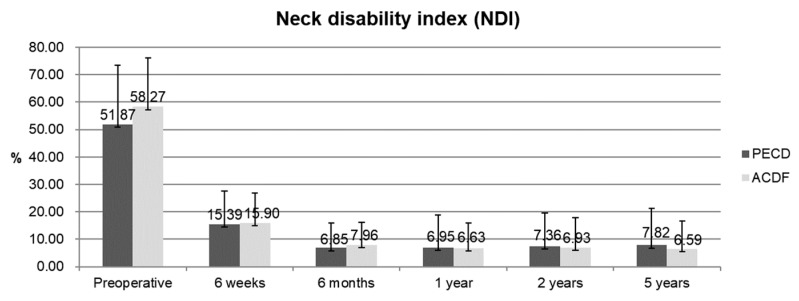
The Neck Disability Index (NDI) preoperatively and at six weeks, six months, one year, two years, and five years postoperatively. There were no significant differences between the groups. PECD, percutaneous endoscopic cervical discectomy; ACDF, anterior cervical discectomy and fusion.

**Figure 4 jcm-09-00371-f004:**
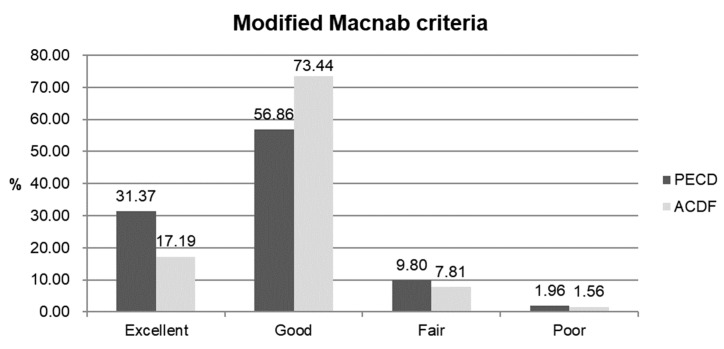
Global outcomes according to the modified Macnab criteria. An excellent or good outcome was observed in 45 of 51 patients (88.24%) in the PECD group and in 58 of 64 patients (90.63%) in the ACDF group. There were no significant differences between the groups. PECD, percutaneous endoscopic cervical discectomy; ACDF, anterior cervical discectomy and fusion.

**Figure 5 jcm-09-00371-f005:**
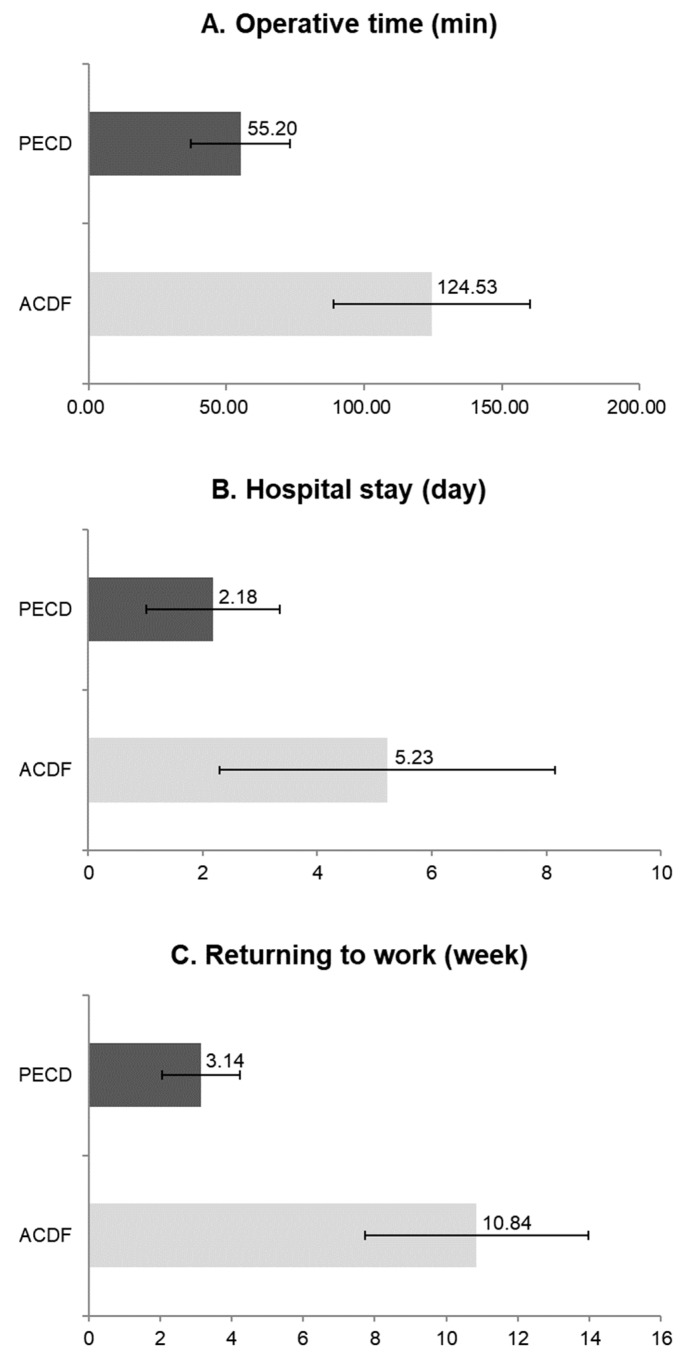
Perioperative data. **A**. Operative time was significantly shorter in the percutaneous endoscopic cervical discectomy (PECD) group than in the anterior cervical discectomy and fusion (ACDF) group (55.20 ± 18.03 min vs. 124.53 ± 35.68 min; *p* < 0.01). **B**. The hospital stay was significantly shorter in the PECD group than in the ACDF group (2.18 ± 1.16 d vs. 5.23 ± 2.93 d; *p* < 0.01). **C**. The time to return to work was significantly shorter in the PECD group than in the ACDF group (3.14 ± 1.08 weeks vs. 10.84 ± 3.12 weeks; *p* < 0.01).

**Table 1 jcm-09-00371-t001:** Demographic characteristics.

	PECD	ACDF	*p* Value
Number of Patients	51	64	
Sex (M:F)	28:23	36:28	1
Mean Age (Years)	42.2 (25–67)	47.5 (20–75)	0.083
Mean BMI (kg/m^2^)	24.06	23.71	0.535
Operative Level			0.273
C3–4	3	5	
C4–5	10	6	
C5–6	22	37	
C6–7	16	16	
Motor Deficit			0.682
Normal	30 (58.8%)	34 (53.1%)	
Mild (Grade 4)	16 (31.4%)	25 (39.1%)	
Moderate (≤Grade 3)	5 (9.8%)	5 (7.8%)	
Sensory Disturbance			0.687
Hypesthesia	8 (15.7%)	12 (18.8%)	
Numbness	32 (69.6%)	39 (60.9%)	

PECD, percutaneous endoscopic cervical discectomy; ACDF, anterior cervical discectomy and fusion; BMI, body mass index; M, male; F, female.

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
