# Peer review of "Percutaneous Endoscopic Cervical Discectomy versus Anterior Cervical Discectomy and Fusion: A Comparative Cohort Study with a Five-Year Follow-Up"

_jcm, 2020, doi:10.3390/jcm9020371_

Round 1

Reviewer 1 Report

I would accept the article for publication.

Reviewer 2 Report

I reviewed again the paper with the authors responses. Although I would like to point out that a multivariate analysis still can be done on retrospective studies to account for confounding, it should not be an absolute contraindication to publishing this manuscript. I recommend accepting the publication at this time. 

Reviewer 3 Report

The authors revised according to reviewers’ requests. This 2nd paper is improved.  I am sure this revised manuscript is ready for publication.

This manuscript is a resubmission of an earlier submission. The following is a list of the peer review reports and author responses from that submission.

Round 1

Reviewer 1 Report

Thank you for this interesting article. I think the authors presented a nice comparison between both groups however the study would benefit from some revisions to make its impact better. Since the study is retrospective, it would be worthwhile including a multivariate analysis to account for known confounders.

Furthermore, it is not clear why a single level ACDF took significantly more time in the OR than what it usually does. Successful ACDFs given the inclusion criteria usually take around 40 minutes to complete. Furthermore, why does a single level ACDF patient stay in the hospital for 5 days?  I do not think this would be the general consensus among spine surgeons. Were there reasons to account for such a prolonged stay? The authors need to explain that.

Reviewer 2 Report

In this study, the authors retrospectively demonstrated the long-term clinical outcomes of percutaneous endoscopic cervical discectomy (PECD) compared with anterior cervical discectomy and fusion (ACDF) for soft cervical disc herniation (CDH). And they concluded that PECD provided the typical benefits of minimally invasive surgery and may be an effective alternative for treating soft CDH through the results that the 5-year outcomes of PECD were comparable to those of conventional ACDF. I guess this report is relatively worthy because of a paucity of the comparative study between the PECD and the conventional ACDF as the authors mentioned.

The methodology of this study was precisely explained. And limitations were also described.

Reviewer 3 Report

Reviewer’s comments on the manuscript entitled “Percutaneous endoscopic cervical discectomy versus anterior cervical discectomy and fusion: a comparative cohort study with 5-year follow-up”.

Authors report on a retrospective review of prospectively collected data of patients with cervical disc herniation and cervical radiculopathy that underwent either anterior cervical discectomy and fusion of percutaneous endoscopic cervical discectomy in a single center during a 2.8y period. A total of 135 patients were included and 20 (14.8%) lost final follow-up at 5 years post-op. Surgical technique was left to surgeon discretion. Follow-up data collection was done at regular visits or telephone interview using VAS and NDI. Authors also included Macnab criteria for global outcome classification. Overall, clinical outcomes were similar in both groups. Data on operative time, hospital stay, and time to return to work favored endoscopic surgery. This retrospective study presents several bias and limitations, most of them acknowledged by the authors in the discussion. Some comments that should be addressed by the authors:

Are the two groups comparable? How was the ACDF group selected for this analysis? Was it done based on image (i.e. only soft discs were selected?), or clinica criteria of cervical radiculopathy treated with ACDF? Highlighting Macnab criteria demonstrates their bias towards endoscopic surgery. How was Macnab criteria defined, by the surgeons? Or by an independent researcher? How did the authors manage missing data? How many patients in each data-point did they analyze? No dispersion measure of continuous variables is reported in the graphs. Box-plot graphs would be more illustrative than bar graphs. First page lines 43-44, the authors state that “very few randomized controlled trials have evaluated the efficacy of PECD (22)”. But only 1 reference is quoted. Are there more studies?